More is not always better: delta-downscaling climate model outputs from 30 to 5-minute resolution has minimal impact on coherence with Late Quaternary proxies

Lucy Timbrell[1,2*,] James Blinkhorn[1,2], Margherita Colucci[1,3], Michela Leonardi[3,4], Manuel Chevalier[5], Andrea Vittorio Pozzi[3], Matt Grove[2], Eleanor Scerri[1,6,7] Andrea Manica[3]

[1] Human Palaeosystems Group, Max Planck Institute of Geoanthropology, Jena, Germany

[2] Department of Archaeology, Classics and Egyptology, University of Liverpool, U.K.

[3] Evolutionary Ecology Group, Department of Zoology, University of Cambridge, Cambridge, U.K.

[4] Natural History Museum, London, U.K.

[5] Meteorology Department, University of Bonn, Bonn, Germany.

[6] Department of Classics and Archaeology, University of Malta, Malta

[7] Department of Prehistoric Archaeology, University of Cologne, Germany

*Correspondence to*: Lucy Timbrell (lucy.timbrell2@liverpool.ac.uk)

**Abstract.** Both proxies and models provide key resources to explore how palaeoenvironmental changes may have impacted diverse biotic communities and cultural processes. While proxies are thought to provide the 'gold standard' in reconstructing the local environment, they only provide point estimates for a limited number of locations. On the other hand, models have the potential to afford more extensive and standardised geographic coverage of multiple bioclimatic variables. A key decision when using model output is the appropriate geographic resolution to adopt; models are coarse scale, in the order of several arc degrees, and so their outputs are usually downscaled to a higher resolution. Most publicly available model time-series have been downscaled to 30 or 60 arc-minutes, but it is unclear whether such resolution is sufficient for certain applications like species distribution models, or whether this may homogenise environments and mask the spatial variability that is often the primary subject of analysis. Here, we explore the impact of increasing the resolution of model output from 30 to 5 arc-minutes using the delta-downscaling method, which interpolates and applies the long-term difference between past and present model datasets to a higher resolution grid of observed present-day climate. We seek to determine to what extent further downscaling captures climatic trends at the site-level through direct comparison with proxy reconstructions, evaluating the different versions of the output from the HadCM3 Global Circulation model for annual temperature, mean temperature of July, and annual precipitation against a large empirical dataset of pollen-based reconstructions from across the Northern Hemisphere. Our results demonstrate that models tend to provide broadly similar accounts of past climate to that obtained from proxy reconstructions, with coherence tending to decline with age and at higher altitudes. However, our results imply that using the delta method to downscale to a very fine resolution has minimal net effect on the coherence of model output with pollen records in most cases. Optimal spatial resolution is therefore likely to be highly dependent on specific research contexts and questions, with careful consideration required regarding the trade-off between highlighting local-scale variations and increasing potential error via unreliable interpolation.

# 1 Introduction

Realistic reconstructions of global paleoclimates are vital for modelling long-term evolutionary and ecological processes in fields like evolutionary biology, palaeoecology, palaeontology, and archaeology. Proxy records, such as those derived from pollen or other biomarkers, tend to be the preferred method for characterising past environments at specific locations; however, in order to extrapolate beyond the individual core sites and across wider regions, often it is necessary to rely on modelled or simulated climatic conditions. Recently, the production of high-resolution simulations, characterising climatic variables across

vast time periods, have allowed for the production and analyses of time series similar to those produced using proxy data (e.g., Fordham *et al.,* 2017; Armstrong *et al.,* 2019; Holden *et al.,* 2019; Beyer *et al.,* 2020; Brown *et al.,* 2020; Karger *et al.,* 2021; Krapp *et al.,* 2021; Timmerman *et al.,* 2022). Openly accessible simulated datasets, such as those published by Beyer *et al.* (2020a), Krapp *et al.* (2021), Yun et al. (2023) and Barreto *et al.* (2023), and associated toolkits (e.g., the analytical package *pastclim* for manipulating and extracting modelled data; Leonardi *et al.,* 2023), are particularly useful for scientists interested

in Pleistocene and Holocene timescales, facilitating continuous-time analyses at a high spatial resolution across a wide range of applications, such as habitat and species distribution modelling (SDM) and the quantitative analysis of climate change in relation to spatiotemporally diverse biological and behavioural phenomena (e.g. Beyer *et al.,* 2021; Padilla-Iglesias *et al.,* 2022; Blinkhorn *et al.,* 2022; Timmerman et al. 2022; Leonardi *et al.,* 2022; Zeller and Timmerman 2024; Mondanaro et al. 2025).


Proxy data, while allowing for detailed reconstructions of climatic conditions through time, are rarely in direct association with archaeological or palaeontological sites, nor do they consistently provide an absolute, linear, and standardised representation of past climate across large geographic areas. In this sense, they often provide relative estimates of past climate, an issue highlighted in a synthesis of eastern African Late-Middle Pleistocene climate records by Timbrell *et al.* (2022),

demonstrating that different proxy records – even from within a relatively spatiotemporally restricted region – can provide alternate ideas of relative 'humidity'. This is the result of the diverse nature of the data employed (i.e., pollen, lake sediments, ice cores etc.), which record climate in an inconsistent way that typically cannot be articulated as the bioclimatic indicators and environmental parameters that are routinely in species distribution models (SDMs) (e.g. Beyer *et al.* 2021; Blinkhorn *et al.* 2022; Leonardi *et al.* 2022). Model output have the potential to overcome these shortfalls, providing tangible values for

parameters such as temperature, precipitation, and a range of derived bioclimatic indices (e.g., Hijmans *et al.,* 2005), that are consistent across variables for a more complete account of climatic conditions. Models additionally offer much wider spatial coverage of the landscape that can be directly related to specific study sites and the palaeoclimatic differences between them. However, the integration of modelled climate with proxy data is not straightforward. For example, using simulations at a coarse resolution can produce biases when compared to on-site proxies due to the underlying complexity of the physical landscape,

particularly in coastal and topographically diverse regions (Maraun and Widmann, 2018). Resultant differences can be in the order of several degrees for temperature and tens of percent for precipitation, which could lead to substantially different biome

classifications and estimations of ecologies experienced (Kottek *et al.,* 2006). Such variations can have important implications for the diverse fields employing model output for the reconstruction of past and present species distributions, dispersal and extinction processes, and biogeographic patterns.


High resolution simulations of multiple time slices are often desired by consumers of model output yet difficult to obtain due to computational costs. For example, dynamical downscaling allows for the detailed description of processes in the climatic system and can improve the capturing of localised climatic conditions (Rummukainen, 2016; Strandberg et al., 2023), however this method is rarely applied in fields like palaeoecology and archaeology due to the computational costs, particularly when a

large number of time steps are required. Most of the recently produced time series of palaeoclimate outputs have been downscaled from the native resolution of the models (usually in the order of 2 or 3 arc-degrees) to a higher resolution of 30 arc-minutes using statistical methods (Fordham et al. 2017; Beyer *et al.* 2020a; Krapp *et al.* 2021; Zeller and Timmerman 2024; Mondanaro et al. 2025) as these approaches can be more easily applied to several time periods. Within statistical downscaling, different methods exist to increase the spatial resolution of model simulations; these include the delta method,

generalised additive models (GAMs), and quantile mapping. These are all aimed at minimising biases in models, characterised as differences in statistical distributions between observed and simulated series. Analyses by Beyer *et al.* (2020b) comparing debiased simulation data and empirical reconstructions at 30-minute resolution indicate the effectiveness of the delta method, which generally produced the most accurate simulation, though with substantial spatial and temporal variation in model performance. To debias simulations, delta-downscaling uses a map of local differences between observed and modelled values

in the present day to correct for bias in the past (Maraun and Widmann, 2018). In this sense, the method assumes that biases are location specific and constant over time. Delta-downscaling can account for some climatic variations in relation to the underlying landscape, such as capturing some of the effects of topography on temperature and rainfall, which can be useful in certain analyses of past processes and dynamics.

As a community, we are becoming increasingly aware of issues related to the scale and resolution of climate variables, yet it is currently unclear what level of downscaling is desirable for applications like SDM. Indeed, the ODMAP (Overview, Data, Model, Assessment, Prediction) protocol stresses the importance of spatial resolution and extent of environmental predictors, as well as processing and scaling (Fitzpatrick et al. 2021), yet there is still no universally agreed upon pipeline for SDM to help determine when downscaling may be important. Recently a resolution of 1km was obtained for the TRACE21K

simulations using the CHELSA algorithm (Karger *et al.* 2023), interpolating very high-resolution climate for every 100 years for the last 21,000 years. Some studies support that much finer-scale simulations have higher predictive power in SDMs of modern populations (Chauvier *et al*. 2022; Ozdemir 2024), though whether such accuracy can be extended to predicted distributions in the past or future is unclear, particularly due to the assumptions of the delta-downscaling method that local biases remain constant through time (Franklin *et al.* 2015). Proxies offer a more localised account of climate in certain places, yet they too can be associated with high degrees of uncertainty, arising from multiple sources. Nonetheless, determining model


agreement with empirical reconstructions from proxies remains a widely applied method for ground-truthing downscaled climatic output.

Previous studies have produced varied results when comparing the climatic time series produced by model simulations with proxy-based reconstructions. Some find that simulations and reconstructions reproduce similar major changes in temperature at large spatial scales (Fernándex-Donado *et al.,* 2013; Zhu *et al.,* 2019), whilst others suggest divergence (Laepple and Huybers 2014; Rehfeld *et al.,* 2018). A recent meta-analysis by Laepple *et al.* (2023) found that studies in the Northern Hemisphere (where data are more abundant) have mixed results, suggesting potential areas of mismatch at local and regional scales. These authors suggest that shortcomings in both model simulations and proxy reconstructions may contribute to this divergence with models being less efficient at simulating local and regional temperature variability at relatively long timescales and methods of temperature reconstruction from proxies facing systematic deficiencies, though stronger emphasis is placed on the former. Strandberg *et al.* (2022) conversely suggest that comparisons between models and proxies are mostly limited by the large errors associated with proxy data.

Given the ever-increasing demand to produce more accurate models of past climate across extended timeframes, we tested whether downscaling climatic models from a relatively coarser (30-min) to a higher resolution (5-min) leads to increased agreement with empirical reconstructions of past climate from proxies. We applied a new suite of functions in the *pastclim* R package (Leonardi *et al.,* 2023) for delta-downscaling model output, including model-data comparisons with directly downscaled HadCM3 outputs from Huntley et al (2023), which is an updated version of that used to generate Beyer et al. (2020a), as well as the model time series from Beyer et al. (2020a). We have provided an assessment of the 2,592 Northern Hemisphere records for the last 30,000 years available from LegacyClimate 1.0 (Herzschuh *et al.* 2023), a pollen-based database reconstructing past annual temperature and precipitation and July temperature, that can be directly compared to variables from these model outputs at varying spatial resolution. Our work has quantified the average divergence between the time series produced using modelled climate at varied spatial resolution and method of proxy reconstruction, with our results ultimately endorsing the use of model output in the absence of high-resolution proxies, though with careful consideration as to the most appropriate resolution for analysis.

## 2 Materials and methods

### 2.1 Climate models

To test the impact of delta-downscaling at different resolutions, we used two time series of model simulations. The first one is a set of raw temperature and precipitation outputs from the HadCM3 Global Circulation Model, at their native resolution of 3.275 x2.5 arc-degrees taken from Huntley *et al.* (2023). We consider a set of simulations in which the HadCM3 was run with appropriate boundary conditions for the last 120k years at 2,00 years intervals (the original set in that paper covered the last

800k years). The second model series comes from Beyer et al. (2020a) within the *pastclim* R package (Leonardi et al. 2023). This is based on an older series of runs of the HadCM3 Global Circulation Model (Singarayer and Valdes 2010, Singarayer and Burrough, 2015; Valdes et al. 2017) for the last 120k years, in 72 snapshots (2,000-year time steps between 120,000 BP and 22,000 BP; 1,000-year time steps between 22,000 BP and the pre-industrial modern era). As in the other set, the original model output of HadCM3 had a grid resolution of 3.75 x 2.5 arc-degrees.

These outputs were first downscaled using a series of runs of the higher resolution HadAM3H model, available at 1.25 x 0.83 arc-degrees for the last 21,000 years in 9 snapshots (2,000-year time steps between 12,000 BP and 6,000 BP; 3,000-year time steps otherwise) using an approached termed dynamic delta downscaling by Beyer et al (2020a). This method consists of generating a set of delta matrices based on the few time steps for which outputs were available from both HadCM3 and HadAM3H, and then using these matrices to downscale each time step in the full set by using a weighted interpolation of the two closest delta matrices based on $CO_2$ (see Beyer et al. 2020a, for details). This approach takes advantage of the higher resolution of local dynamics captured by HadAM3H, which is computationally too expensive to be run for all time steps. These outputs were then debiased and downscaled in Beyer et al. (2020a) to 0.5 x 0.5 arc-degrees with the delta method using the Climate Research Unit Global Climate Dataset (CRU) as the modern climatic reference (Mitchell and Jones, 2005).

For this study, we delta downscaled and debiased these two model outputs to a resolution of both 30 arc-minutes and 5 arc-minutes using modern observation from WorldClim2 (Fick and Hijmans, 2017). For the Beyer et al (2020a) model, as it was already at 30 arc-minutes, the delta downscaling at this resolution gives us a debiased version based on WorldClim2 rather than CRU. We used a global relief map from ETOPO2022 (NOAA National Center for Environmental Information, 2022) to reconstruct past coastlines following sea level change (Spratt and Lisiecki, 2016). We selected WorldClim2 as the modern reference as the transfer functions used in the LegacyClimate1.0 dataset were also derived from this dataset (at 30-minute resolution), allowing us to control for the effects of the modern data used for debiasing on our results. All data manipulations were done using the R package *pastclim* (Leonardi et al. 2023).

Downscaling was performed one monthly variable at a time (i.e., January temperature) by taking the coarse simulations from Beyer *et al.* (2020a) and Huntley et al. (2023) with the corresponding set of high-resolution modern simulations from WorldClim2 (Fick and Hijmans, 2017) and equally high-resolution global relief map (NOAA National Centres for Environmental Information, 2022). Through integrating both bathymetric and topographic values for masking sea level changes, a delta raster was computed, adding the difference between past and present-day simulated climate to present-day observed climate, following Beyer et al. (2020a) and Krapp et al. (2021) The delta method therefore assumes that local (i.e. grid-cell-specific) model biases are constant over time (Maraun and Widmann, 2018). The resulting matrix only covers the land extent at the present. We then expanded this matrix to reach the largest land-extent in any of the times-steps under consideration using an inverse-distance-weighted interpolation. For most of the world, at the resolution of 30 and 5 arc-

minutes, this only requires interpolating a small number of cells away from the coastline; for higher resolutions, other interpolating algorithms might be more appropriate. We note that the delta-downscaling can also be obtained by creating first the difference between model outputs, which is then applied to the observational model. However, such a direction is more

computationally expensive, as the interpolation outside the coastlines would have to be repeated for each time step.

For temperature variables, the bias in a geographical location $x$ (a cell with a given latitude and longitude) is given by the difference between present-day observed $T_{obs}(x, 0)$ and simulated $T_{sim}^{\oplus}(x, 0)$ temperature, interpolated to the desired higher resolution grid via bilinear interpolation. Downscaled temperature ($T_{sim}^{DD}$) in $x$ at time $t$ is thus estimated as

$$T_{sim}^{DD}(x,t) := T_{sim}^{\oplus}(x,t) + \left(T_{obs}(x,0) - T_{sim}^{\oplus}(x,0)\right)$$

(1)

Precipitation is lower bounded by zero and covers different orders of magnitude across different regions compared to

180 temperature. Multiplying rather than adding the bias correction is common when applying the delta method for precipitation, which corresponds to applying the simulated relative change to the observations (Maraun and Widmann, 2018). However, this method can therefore be hypersensitive in drylands, leading to overprediction of precipitation (and thus exacerbating the 'drizzling' bias of GCM). We have therefore adopted an additive approach for precipitation, analogous to the one used for temperature, with clamping within the range of observed maximum and minimum for current climate (Beyer et al. 2020a;

Huntley et al. 2023). Like temperature, downscaled precipitation is estimated as

$$P_{sim}^{DD}(x,t) := P_{sim}^{\oplus}(x,t) + \left(P_{obs}(x,0) - P_{sim}^{\oplus}(x,0)\right)$$

(2)

The resulting monthly datasets were then utilised within the *pastclim* framework to recompute the 17 bioclimatic variables available in the original dataset (Supplementary Table S1), with mean annual temperature (bio01), mean temperature of the warmest quarter (bio10) and total annual precipitation (bio12) extracted here for further analysis given their relevance to the variables captured by the proxy reconstructions employed.

Interpolating over small spatial extents can lead to the introduction of artefacts due to the application of inverse distance weighted interpolation, which takes information from neighbouring cells to produce high-resolution reconstructions (Beyer *et al.* 2020b). Given the wide spatial distribution of the proxy dataset, we thus performed downscaling for the entire world for all of the time steps available in Beyer et al. (2020a) and the HadCM3 GCM (Huntley *et al.* 2023) for the last 120,000 years. The global downscaled bioclimatic variables have been made available on Zenodo (https://doi.org/10.5281/zenodo.7828453) for

future use. Figure 1 shows the different climatic models tested in this research for both the present day and the Last Glacial Maximum (LGM) and the geographic coverage of the proxy records.

## HadCM3 30-min model (WC) - present day

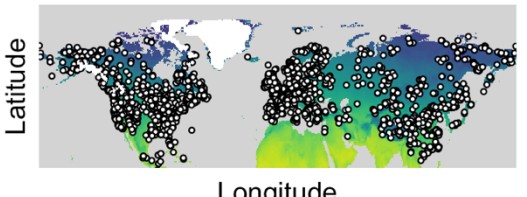

## HadCM3 30-min model (WC) - LGM

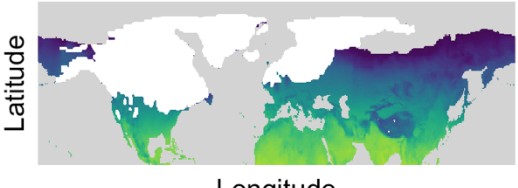

## HadCM3 5-min model (WC) - present day

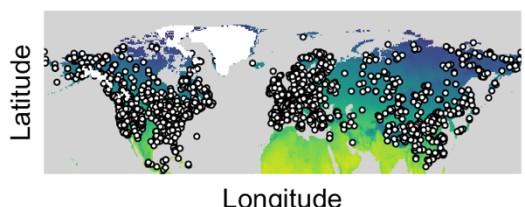

## HadCM3 5-min model (WC) - LGM

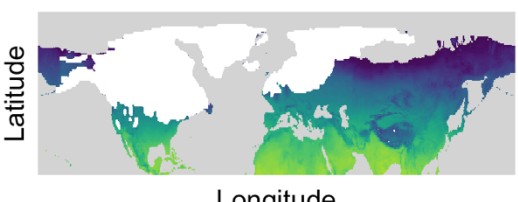

## Beyer 30-min model (CRU) - present day

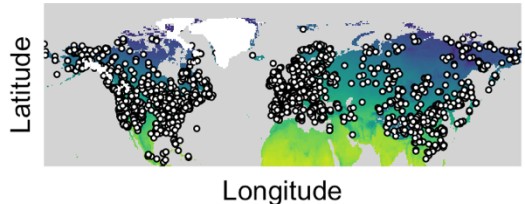

## Beyer 30-min model (CRU) - LGM

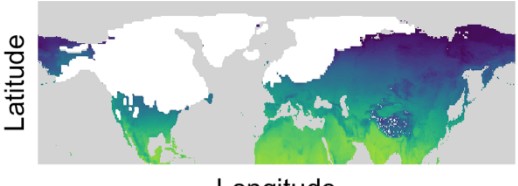

## Beyer 30-min model (WC) - present day

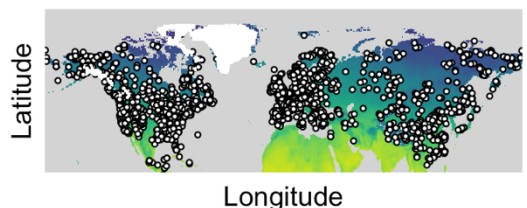

## Beyer 30-min model (WC)- LGM

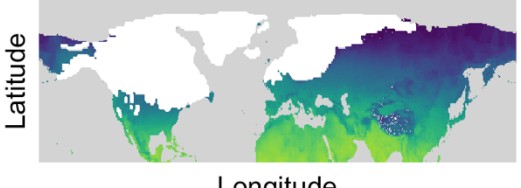

## Beyer 5-min model (WC) - present day

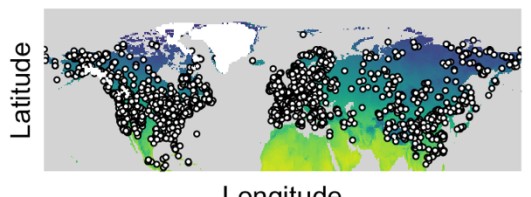

## Beyer 5-min model (WC) - LGM

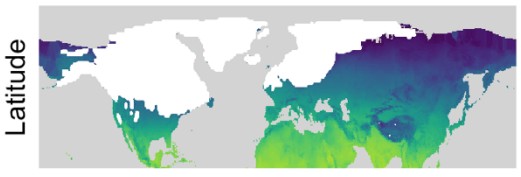

Figure 1. Site locations of proxy records studied in this analysis (left), against mean annual temperature (bio01) from the different model outputs for the present day and the Last Glacial Maximum (LGM), manipulated within pastclim (Leonardi *et al.* 2023). Land mass in each time slice is masked by global ice sheets (plotted in white) and predicted sea level.

## 2.2 Proxy reconstructions

We employed the LegacyClimate 1.0 proxy dataset by Herzschuh *et al.* (2023) for direct validation of the model outputs. Mean annual temperature ($T_{ann}$), Mean July temperature ($T_{july}$) and total annual precipitation ($P_{ann}$) were reconstructed from fossil pollen data using the Weighted-Averaging Partial Least Squares (WA-PLS) and Modern Analogue Technique (MAT) methods, both of which are widely used and generate similar time series, though each method's performance vary in response to various factors, such as the quality and diversity of the calibration data, the time interval to be reconstructed, and the resolution of the pollen data (Sweeney *et al.,* 2018; Birks *et al.* 2010; Chevalier *et al.,* 2020). In LegacyClimate 1.0, the diverse pollen records are handled consistently through merging taxa into high-level harmonised taxonomic groups, increasing the possibility of matching modern climate analogues and fossil datasets. Its geographic coverage across the Northern Hemisphere is also much larger than other databases (e.g. Mauri *et al.,* 2015; Marsicek *et al.,* 2018; Routson *et al.,* 2019). Our use of a single database reconstructing climate based on a single proxy reduces inter-site variability resulting from the type of data utilised and allows the generation of analogous climatic parameters with direct relevance to bioclimatic variables available in the Beyer *et al.* (2020a) model; $T_{ann}$, $T_{july}$ and $P_{ann}$ from LegacyClimate1.0 are the equivalent bioclimatic variables to bio01, bio10 and bio12 from HadCM3 GCM (Huntley *et al.* 2023) and Beyer et al. (2020a) model time series, which are standardly used in climatic modelling.

To facilitate comparison between the proxy reconstructions and the model outputs, we interpolate each proxy record via bilinear interpolation to the equivalent chronological resolution of the climatic models to enable quantification of differences between the time series; interpolating to regular time intervals ensures that periods of particularly dense sampling in the original cores do not exert undue influence on the results. For this, we extracted the climatic values from the model at the coordinates of the proxy site for the time steps captured in the proxy record. Following data-cleaning, we retain 2,385 records from LegacyClimate1.0. One record was removed as it did not have any proxy data associated with the MAT method (ID Dataset: 100127), a further 25 were omitted due to a lack of consistent time steps in the models being available, and an additional 170 records were removed as they fall under the cropped sea-level of the models. The latter includes some proxy sites that are located on small islands not captured by the model or within lake margins. Table 1 summarises the proxy records and climatic model outputs studied in this research.

Table 1. Summary of the proxy records selected from the LegacyClimate 1.0 (Herzschuh *et al.,* 2023) and the model outputs (Beyer *et al.,* 2020a; Huntley et al. 2023) selected for analysis of mean annual temperature (bio01, $T_{ann}$), mean July temperature (bio10, $T_{july}$) and total annual precipitation (bio12, $P_{ann}$).

| | Regions | N / Cell size | Type of data | Climatic variables extracted | Time min (1,000 years ago) | Time max (1,000 years ago) | Mean freq of records (years) | Reference (and doi) |
|---|---|---|---|---|---|---|---|---|
| Legacy Climate 1.0 | Asia East North America West North America Europe | 2385 proxy sites | Pollen reconstructions | $T_{ann}$ $T_{July}$ $P_{ann}$ | 0 | 30 | 670 | Herzschuh, U. *et al.* (2023). *Scientific Data* (10.5194/essd-15-2235-2023) |
| HadCM3 Global Circulation model | Global | 30- and 5-min grid cells | Simulations, de-biased and downscaled using WordClim2 (this paper) | Bio01 Bio10 Bio12 | 0 | 120 | 1,000 until 23,000 years ago and then every 4,000 years | Huntley, B. et al. (2023). *Journal of Biogeography*, (10.1111/jbi.14619) |
| Beyer *et al.* (2020a) statistics-based simulations | Global | 30- and 5-min grid cells | Simulations, de-biased and downscaled using CRU (original) and WordClim2 (this paper) | Bio01 Bio10 Bio12 | 0 | 120 | 1,000/2,000 | Beyer, R. *et al.* (2020a). Scientific Data (10.1038/s41597-020-0552-1) |

## 2.3 Analysis

To quantify the differences between time series, we calculated the bias, root mean square error (RMSE) and normalised RMSE (NRMSE). The RMSE measures the coherence between the model simulations and the proxy reconstructions, whilst the bias (calculated as the mean residual) highlights whether the coarse or downscaled model overestimates (positive values) or underestimates (negative values) the proxy records. Standardising the RMSE using the mean allows comparing the coherence between variables. The bias can also be considered per proxy record to show which areas are over or underestimated for any

given variable, facilitating comparability. Considering that downscaling to higher resolutions is thought to capture spatial variations in climate, we tested the statistical significance of differences in model-data coherence between lower resolution (30-min) and higher resolution (5-min) models, using a standard significance threshold of $p < 0.05$ via the Kruskal-Wallis non-parametric test. We also calculated the proportion of proxy records (reconstructed using the MAT and WA-PLS methods) that show higher RMSE with 30-min models compared to 5-min models. Instances where the proportion is higher than 0.5 highlight

a positive net effect of downscaling on model-data coherence.

These analyses allow us to evaluate the coherence between the output of the climate models and the reconstructions of specific climatic parameters from proxy data, depending on geographic region, Marine Isotope Stage (chronology), method of climate reconstruction employed in the proxy datasets (MAT versus WA-PLS), elevation of site location (with sites above 1500 meter

above sea level analysed as a subset) and topographic roughness (defined as the energetic cost of movement, see SOM. 1), with areas that require over 200 joules per meter to transverse deemed to have 'high roughness'). All these factors could potentially impact the articulation between the climatic model outputs and the proxy records.

## 3 Results

Figure 2 highlights a sample of non-interpolated time series from proxy sites across the geographic span of the LegacyClim1.0

dataset, demonstrating the coherence through time between different models and empirical reconstructions (WA-PLS and MAT) of the three climatic parameters (annual temperature, July temperature and annual precipitation). Our results show that overall proxy reconstructions and model simulations tend to highlight very similar climatic trends across variables, with average bias across all comparisons for both annual and July temperature time series falling under 1 degree Celsius and annual precipitation less than 40 mm (Fig. 2, Appendix A Tables A1-3). Considering the NRMSE, the most divergent variable on

average is mean annual temperature, particularly for the output of the HadCM3 30-min model (Appendix A Tables A1-3). This result contrasts with other large-scale studies (Bartlein et al. 2011; Chevalier et al. 2021), potentially due to the assumptions made for the proxy reconstructions employed that modern analogues should be utilised from within 2000km around each site. Precipitation should be less affected given that it is more variable through space however temperature tends to be much more autocorrelated, meaning that much colder/warmer temperatures occurring in the past may not occur within

these geographic limits. We find that time series of annual precipitation and July temperature show consistently lower NRMSE

values than mean annual temperature across our model-data comparisons (Appendix A Tables A1-3). These two variables also show highly comparable results between different versions of the model outputs, even at varying spatial resolution and when using different modern reference datasets for downscaling (Appendix A Tables A2-3). The output from the Beyer et al. (2020a) 30-min model (CRU) shows the most consistent net positive effect of downscaling (Supplementary Table S1), probably due
to the difference in modern reference data used for debiasing. However, the overall difference in coherence between the two resolutions of both outputs is judged as minimal for all three variables, particularly when controlling for the modern dataset (Appendix A Tables A1-3), as none of the subsets of model-data comparisons highlighted statistically significant differences between models at 30-min and 5-min resolution (Supplementary Table S1).

Our results based on all of the comparisons in the dataset highlight that the 30-min model time series of annual temperature from Beyer *et al* (2020a) debiased using CRU as the modern reference tends to estimate slightly lower temperatures than those produced by proxy reconstructions (as highlighted in the negative bias results reported in Appendix A Table A1). All other model outputs de-biased using WorldClim2 (WC) at both 30- and 5-min resolution contrastingly tend to predict higher annual temperatures compared to proxy records. For the HadCM3 model output, the model-data coherence is not significantly
different between the 30-min and 5-min model, with less than half of the proxy records seeing improvement in coherence in the 5-min model (49% MAT method, $p = 0.4904$:  46% WA-PLS method, $p = 0.4961$; Supplementary Table S1). Similarly, annual temperature time series from the Beyer et al. (2020a) 30-min (CRU) simulations tend to have more error in only around half the records compared to the higher resolution version, at 51% (MAT method, $p = 0.4904$) and 50% (WA-PLS method, $p = 0.4961$) of proxy sites, with the Beyer et al. (2020a) 30-min (WC) having more error in slightly less than half of records
compared to the Beyer et al. (2020a) 5-min model, at only 49% (MAT method, $p = 0.4904$) and 47% (WA-PLS method, $p = 0.4961$) (Supplementary Table S1).

Whether models tend to predict higher or lower precipitation compared to proxy reconstructions varies for different subsets of the data, though negative bias is particularly prominent in the 30-min model outputs compared to the 5-min equivalents
(Appendix A Table A2). However, again, the overall difference in performance between the two resolutions is marginal for both model time series. Model data-coherence for annual precipitation is not significantly different between the 30-min and 5-min HadCM3 model outputs, with less than half of the records (49%) returning higher RMSE at the coarser resolution (MAT and WA-PLS method, $p = 0.4943$ and $p = 0.4961$; Supplementary Table S1).  Annual precipitation time series from the Beyer et al. (2020a) 30-min model (CRU) have more error in 55% of records (MAT method and WA-PLS methods, $p = 0.4923$ and
$p = 0.4961$ respectively) than the higher resolution version (Supplementary Table S1), whereas the Beyer et al. (2020a) 30-min model (WC) shows higher RMSE in 48%  of time series (MAT and WA-PLS methods, $p = 0.4936$ and $p = 4961$) (Supplementary Table S1).

Models of mean temperature of warmest quarter almost always slightly underestimate temperatures compared to proxy

reconstructions of mean July temperature, regardless of resolution (Appendix A Table A3). This could be linked to the slight

discrepancy in the climatic parameter being captured between the models and the proxies. Average difference in model-data

coherence between the two spatial resolutions is not statistically significant for either the HadCM3 or the Beyer et al. (2020a)

model output, with the July temperature time series from the Beyer et al. (2020a) 30-min model (CRU) showing less coherence

in 58% (MAT method, p = 0.4904) and 56% (WA-PLS method, p = 0.4961) of proxy reconstructions when compared to that

from the Beyer et al. (2020a) 5-min model (WC), although again the Beyer et al. (2020a) 30-min model (WC) shows higher

error in less than half of the proxies  (47%, MAT method, p = 0.4904, WA-PLS method, p = 0.4961) (Supplementary Table

S1). Results for the HadCM3 output mirror closely that of WC-debiased Beyer et al. (2020a) models (49% for the MAT

method, p = 0.4904, and 47% for the WA-PLS method, p = 0.4961).


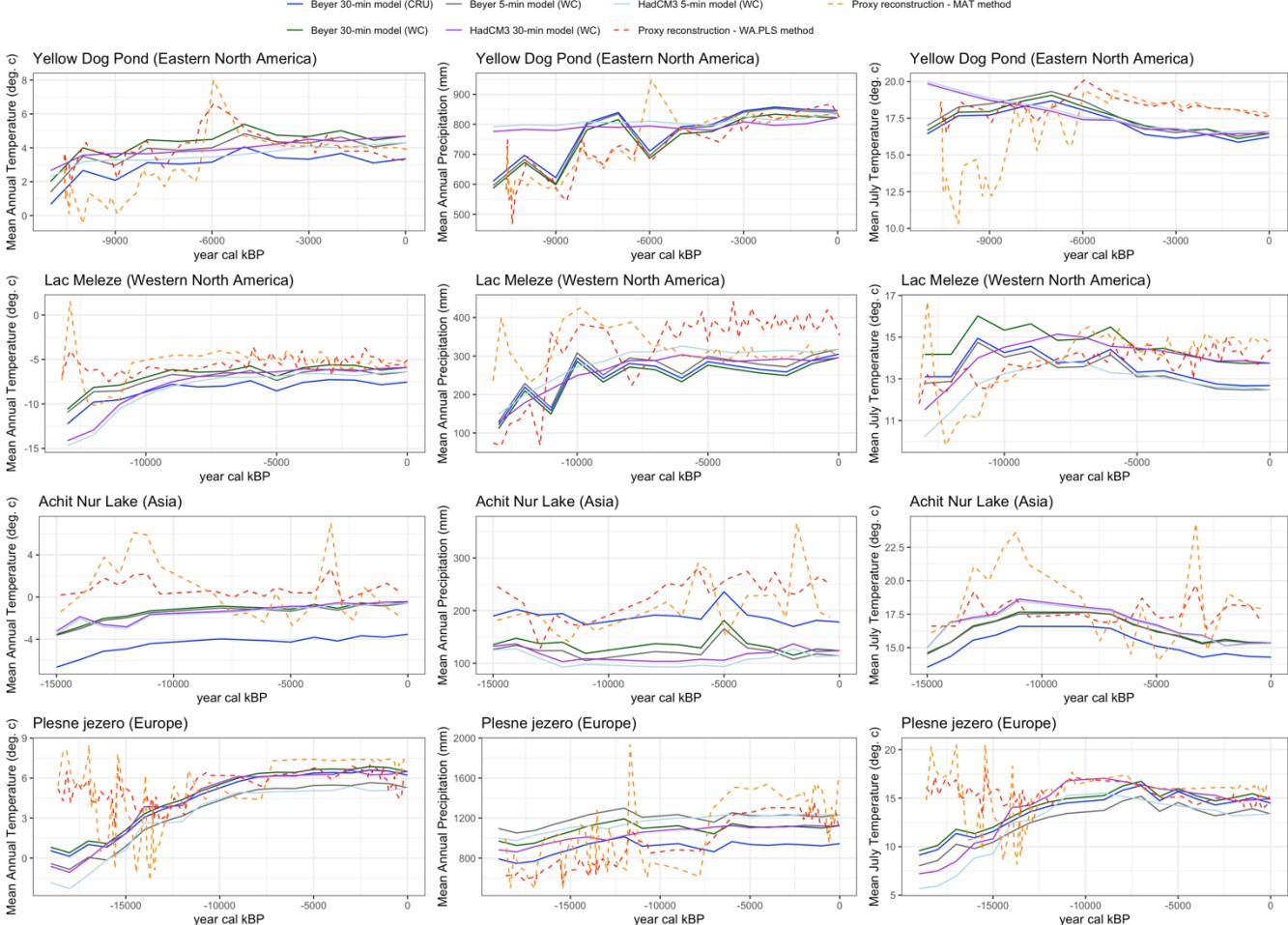

Figure 2. A sample from each regional group of simulated mean annual temperature (left), mean July temperature (middle) and total annual precipitation (right) time series, comparing different model outputs (solid lines) and corresponding non-interpolated proxy reconstructions from LegacyClimate 1.0 (Herzshuch *et al.,* 2021) (dashed lines).

## 3.1 Regional differences

As highlighted in Fig. 3 and Supplementary Fig. S1-2, our results demonstrate some key differences between regions. Firstly, for annual temperature, average bias in European records is positive, suggesting model output in this region tends to overestimate temperatures compared to proxy records, whereas for all other regions annual temperature bias is negative (Appendix A Table A1). Asia and Europe have the highest NRMSE (indicating the highest divergence between proxy records and model outputs) for annual temperature in the Beyer et al. (2020a) 30-min model output (CRU) (Appendix A Table A1,

Supplementary Fig. S1). However, Asia sees higher model-data coherence in both types of 30-min (WC) model outputs compared to their equivalent downscaled 5-min (WC) outputs, whereas the HadCM3 30-min model output produces very high NRMSE for European records (Appendix A Table A1; Supplementary Fig. S1). Downscaling the HadCM3 model output for annual temperature to a 5-min resolution has a positive impact on average coherence in Europe (Appendix A Table A1; Supplementary Figure S1), although this effect is reflected in less than half of the pair-wise comparisons (Supplementary Table

S1). In East North America, average model-data coherence is improved by downscaling in the HadCM3 model output for annual temperature, however the Beyer et al. (2020a) 5-min model output has higher NRMSE than the equivalent 30-min model outputs (Appendix A Table A1; Supplementary Fig. S1). In West North America, the Beyer et al. (2020a) 30-min (WC) and 5-min (WC), as well as the HadCM3 5-min (WC), model outputs for annual temperature are more coherent with the proxy records than the Beyer et al. (2020a) 30-min (CRU) model and the HadCM3 3-min (WC) model outputs, with little difference

between the two resolutions for the Beyer et al. (2020a) model debiased with WC (Appendix A Table A1; Supplementary Fig. S1).

Average model-data bias for precipitation varies regionally, with Europe, West North America and East North America showing consistently negative bias, suggesting that the models underestimate rainfall in these regions (Appendix A Table A1;

Supplementary Fig. S1), in contrast to Asian localities where often average precipitation bias is positive. Model-data coherence for precipitation is highly similar across different resolutions of model output debiased using WC for East North America and Europe, whereas Asia and West North America has less coherence with proxy records in Beyer et al. (2020a) CRU 30-min model and the Beyer et al. (2020a) 5-min model (Appendix A Table A2; Supplementary Fig. 1). Precipitation proxy reconstructions from West North America show the highest NRMSE with the HadCM3 outputs, whereas for Asia the highest

NRMSE model-data comparison is the Beyer et al. (2020a) CRU model, followed by the HadCM3 outputs (Appendix A Table A2; Supplementary Fig. S1)

July temperatures have negative model-data bias for all regions except in Asia for the WC-debiased Beyer et al. (2020a) 30-min and 5-min model output (Appendix A Table A3; Supplementary Fig. 1). In West North America, NRMSE is higher in the

HadCM3 model outputs compared to that from Beyer et al. (2020a), with no differences between resolutions in the latter (when debiased using WC) and a slight improvement in coherence due to downscaling in the former (Appendix A Table A3;

Supplementary Fig. 1). There is no difference in average NRMSE between resolutions of model output for July temperatures in East North America, apart from the Beyer et al. (2020a) 30-min (CRU) model which has higher model-data divergence (Appendix A Table A3; Supplementary Fig. 1). In Asia, downscaling the Beyer et al. (2020a) 30-min (WC) and the HadCM3 model output improves coherence, whereas in Europe these higher resolution model outputs lead to slight decreases in coherence (Appendix A Table A3; Supplementary Fig. 1).

Fig. 3 and Supplementary Fig. S2 highlight these spatial heterogeneities in bias across the Northern Hemisphere, which could have many potential different sources, i.e. geographic variation in the performance of the model outputs, the quality of the present-day calibration data for LegacyClimate1.0 or the modern reference used for debiasing, and/or the impact of confounding variables on the pollen-climate relationships. The East North American subset of proxy reconstructions appear to be the most coherent with the model outputs, generally showing the lowest NRSME values across all variables (Appendix A Table A1-A3; Supplementary Fig S1.). Europe tends to show the lowest proportion of records where error is higher in the coarser models (30 min) compared to the higher-resolution models (5 min), with downscaling having the strongest impact on model-proxy divergence in East and North America, particularly when compared to the Beyer et al. (2020a) 30-min model (CRU) (Supplementary Table S1). Regions showing the least coherence varies depending on the climatic parameter, with Asia and East North America having the highest RMSE values for annual temperatures (Appendix A Table A1; Supplementary Fig. S1), Asia and West North America for precipitation (Appendix A Table A2; Supplementary Fig. S1) and East North America for July temperatures (Appendix A Table A3; Supplementary Fig. S1). Overall, no region shows a statistically significant difference in model-data coherence between models of different resolutions (Supplementary Table S1 and Figure S1). Indeed, often the coarser models have a higher proportion of proxy records with lower error than the 5-min models (Supplementary Table S1), particularly in Europe and Asia, suggesting higher resolutions could simply be adding noise in many scenarios.

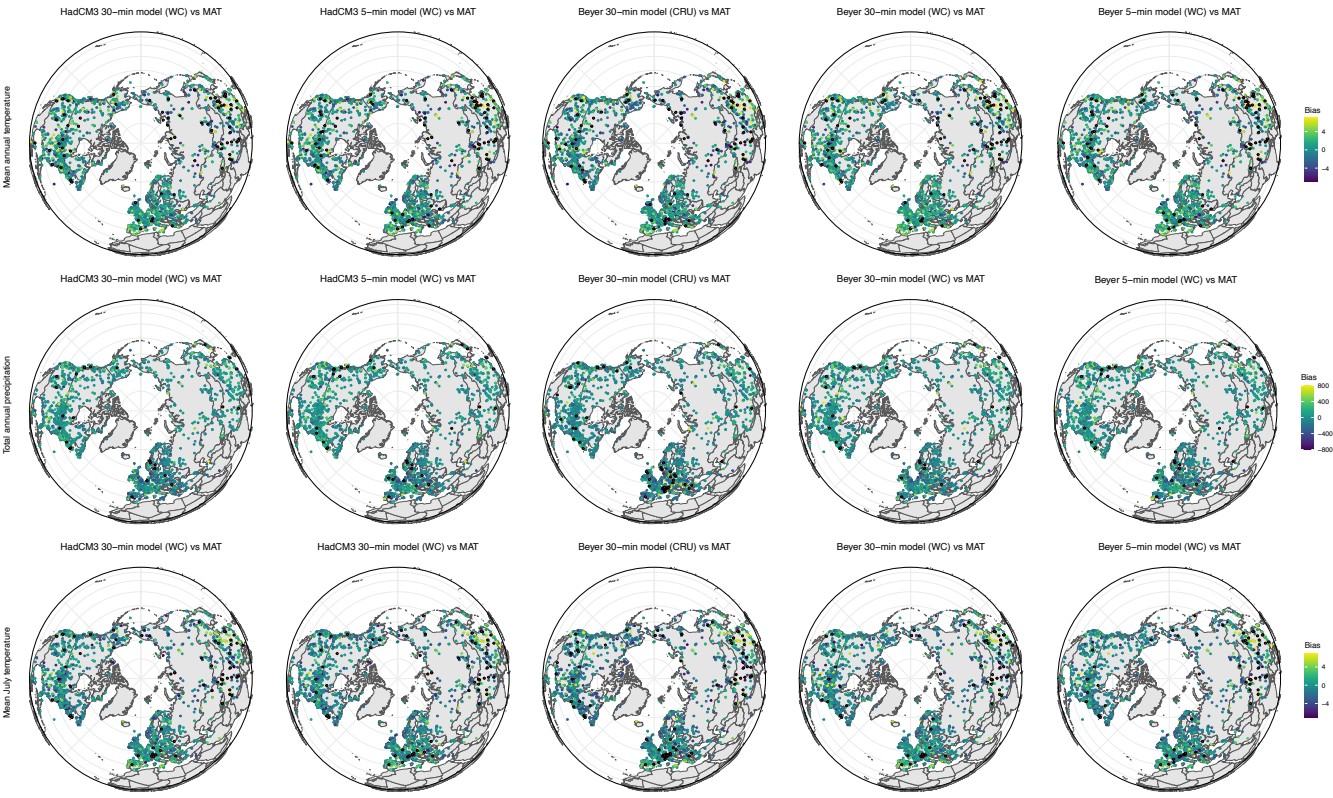

Figure 3. Absolute bias for mean annual temperature, mean annual precipitation, and mean July temperature for each proxy site, comparing the climatic values produced by the MAT method of proxy reconstruction against different versions of the HadCM3 GCM and Beyer *et al.* (2020a) model. Outliers have been highlighted in red, defined as =< -5 and => 5 degrees Celsius for mean annual temperature and July temperature, and =< -800 and => 800 millimetres for total annual precipitation. Visualisation of bias for the WA-PLS method is reported in Supplementary Figure S2.

**3.2 Effects of landscape heterogeneity**

Downscaling model outputs to a very high resolution is often performed to account for smaller-scale landscape features that can locally impact climatic conditions, such as topography and coastlines (Fig. 4). Figure 4 highlights these effects of increasing model resolution in different areas of varying landscape complexity; for example, in the Pittsburg Basin (which is inland and flat) there is little change in the climate signal captured at proxy sites (white circles) following downscaling, whereas, in southern Italy and the Qillian Mountains, downscaling captures more localised details in climates associated with landscape-level variations. Proxy records at higher elevations and topographic complexity may therefore be expected to show stronger coherence with the higher resolution models compared to those at relatively lower resolution.

However, our analysis presents mixed results; for example, for annual temperature, subsets of proxy records at higher altitudes and in regions of higher topographic roughness both have higher NRMSE for the 30-min HadCM3 model compared to the equivalent 5-min version for the MAT method, yet for the WA-PLS method downscaling this output increases NRMSE for records in areas of higher roughness(Appendix A Table A1; Supplementary Fig. S3). Similarly, a negative effect of downscaling on model-data coherence for locations of high roughness is observed for the Beyer et al. (2020a) 30-min model output (WC) for both the MAT and WA-PLS method, as well as proxy reconstructions using the MAT-method in high altitude areas (Appendix A Table A1; Supplementary Figure S3). Annual temperature at higher elevations and topographic complexity modelled based on Beyer et al. (2020a) 30-min (CRU) has consistently higher NRMSE compared to alternate versions of this model output, although the 30-min HadCM3 30-min model is the most divergent from proxy records, particularly for high altitude locations (Appendix A Table A1; Supplementary Fig. S3). In lower altitude and flat locations, downscaling the HadCM3 model shows modest improvements in NRMSE whereas the Beyer et al. (2020a) 5-min (WC) model output is less coherent for these subsets than the equivalent 30-min (WC) version (Appendix A Table A1; Supplementary Fig. S3). In terms of proportions of records that show more error at coarser resolutions, the high-altitude subset consistently has a net positive impact of downscaling for annual temperature, yet no model-data comparisons highlight statistically significant differences in coherence (Supplementary Table S1). Our results also show that proxy reconstructions tend to indicate warmer temperatures at higher elevations and/or in areas of higher topographic roughness compared to model outputs and colder temperatures at lower elevations and/or lower topographic roughness (Appendix A Table A2). This is a known bias of transfer functions when constructing more 'extreme climates' from proxies, given that elevation negatively correlates with temperature and these functions rely on averages of data from modern calibration data sets (Chevalier *et al.,* 2020).

For precipitation, only in low altitude and/or flat areas does the Beyer et al. (2020a) 30-min model (CRU) produce lower values than the proxy reconstructions, indicated by negative bias (Appendix A Table A2; Supplementary Fig. S3). NRME tends to be higher in areas of high altitude (particularly) and areas of high topographic roughness (Appendix A Table A2; Supplementary Fig. S3), however the higher resolution versions of the models do not show an improvement in coherence. For these subsets, the Beyer et al. (2020a) WC model outputs show better average coherence than the Beyer et al. (2020a) CRU and the HadCM3 outputs (Appendix A Table A2; Supplementary Fig. S3). Our results highlight that subsets of low altitude and low roughness proxy records tend to show more instances of downscaling improving the model-data coherence compared to subsets of high altitude and high roughness records, although these are minimal and not statistically significant (Supplementary Table S1, Fig. S3).

Models of July temperatures always produce lower values than that of proxies, regardless of landscape properties (Appendix A Table A3; Supplementary Fig. S3). Our results suggest that, apart from downscaling the HadCM3 model output where minimal improvements in NRMSE are noted, model-data coherence for July temperature is not affected by model resolution when controlling for the modern referenced used to debias (Appendix A Table A3; Supplementary Fig. S3). Overall, we find

that the proportion of proxy records that show higher error (NRMSE) with lower resolution models than higher resolution is around half for all subsets according to landscape variations, indicating no statistically significant effect of further downscaling on data-model coherence, even in areas of landscape heterogeneity (Supplementary Table S1, Fig. S3).

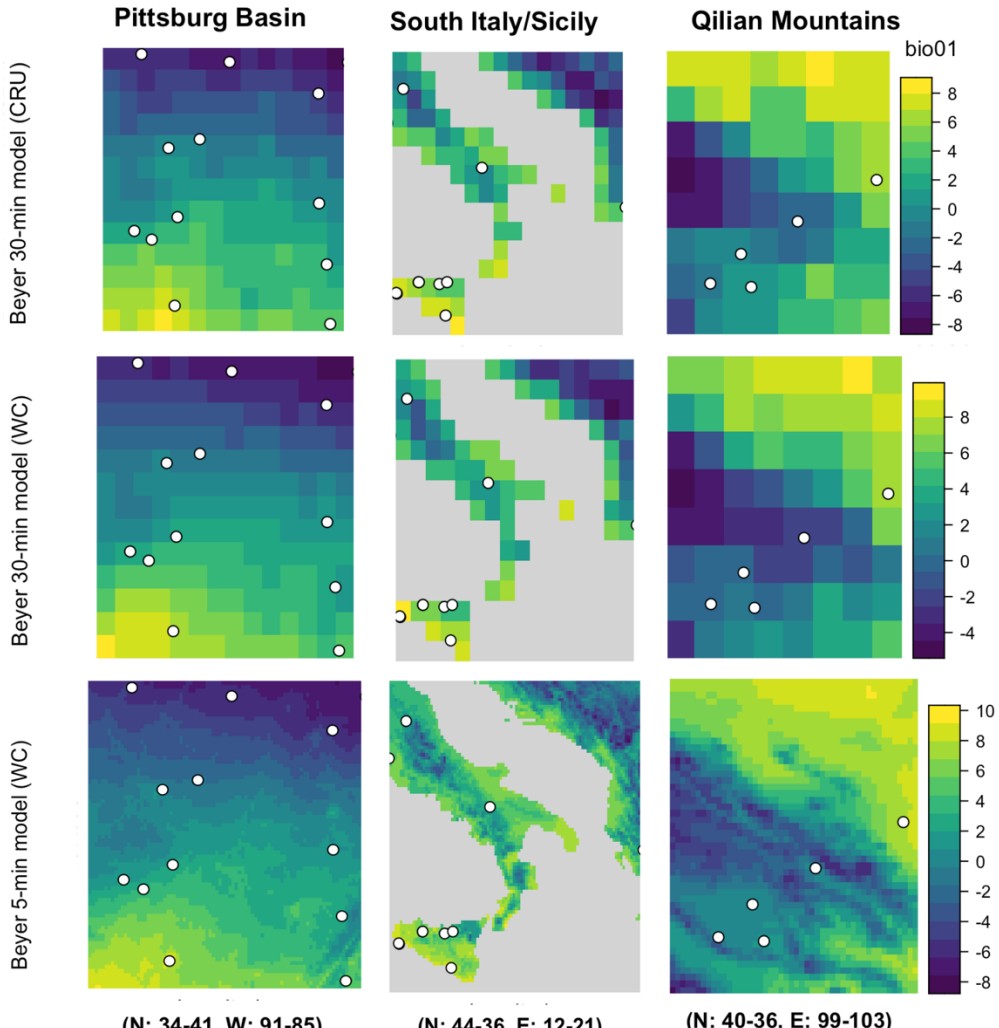

Figure 4. Three regional examples of modelled mean annual temperature for the present day (bio01), demonstrating how downscaling increases spatial resolution by capturing the effects of landscape dynamics through space on climate depending on the underlying topography. Geographic variability in temperature is shown, as simulated by the Beyer *et al.* (2020a) 30-min model output (CRU), Beyer *et al.* (2020a) 30-min model output (WC), and Beyer *et al.* (2020a) 5-min model output (WC), Locations of proxy locations from LegacyClimate 1.0 are shown as white circles.

### 3.3 Glacial versus inter-glacial variability

We then examined discrepancies in model-data coherence through time, separating time slices from the model outputs covering the present day (i.e. timeslice 0), Marine Isotope Stage 1 (MIS 1; 0 – 14,000 years ago) and MIS 2 (14-29,000 years ago). In total, 1060 records were associated with the present day (44% of dataset), 2363 records captured time slices in MIS 1 (99% of dataset) whereas 473 spanned into MIS 2 (20%). Separate analysis of interpolated data points capturing the present day was performed, as the pollen proxies captured in these records should be highly representative of modern ecological communities whilst model data points are based on present-day observations as opposed to simulations into the past, thus providing somewhat of a baseline of model-data divergence.

Our results demonstrate that data points representative of the present have the lowest NRMSE (Appendix Tables A1-3; Fig. 5), though considerable error in some time series exists (Fig. 5). In contrast, the smaller subset of time series covering MIS 2 show the highest bias and NRMSE (Appendix Tables A1-3; Fig. 5), both across model outputs and resolutions, as well as methods of proxy reconstruction. Models capturing older time periods underestimate annual and July temperatures compared to proxy reconstructions and (often) overestimate annual precipitation (Appendix A Tables A1-A3; Fig. 5). We find that the proportion of proxy records that show higher RMSE (and therefore are less coherent) with lower resolution models compared to those of higher resolution is almost always over half for the present day, with annual temperature and July temperature during MIS 2 seeming to also see a slight benefit of downscaling, though this is not statistically significant for any comparison (Supplementary Table S1).

Figure 5 highlights the differences between RMSE values from the present day, MIS 1 and MIS 2, confirming that data-model discrepancies tend to increase with age though not significantly so ($p > 0.05$). Chronological uncertainties in the proxy age model may complicate the comparison between climate simulations and pollen-based records, as well as the process of signal smoothing via interpolation to facilitate analysis. Delta-downscaled models are also inherently designed to replicate current rather than past climate patterns, and proxy reconstructions rely on the identification of modern analogue species that may have a different link to climate than palaeoecological communities, likely further contributing to higher divergence in older time periods (Chevalier *et al.* 2020). Nonetheless, all of the distributions highlighted in Fig. 5 are highly positively skewed even after normalisation– there are many extreme values– confirming that age is just one contributing factor in the divergence between time series (Supplementary Fig. S1, S3).

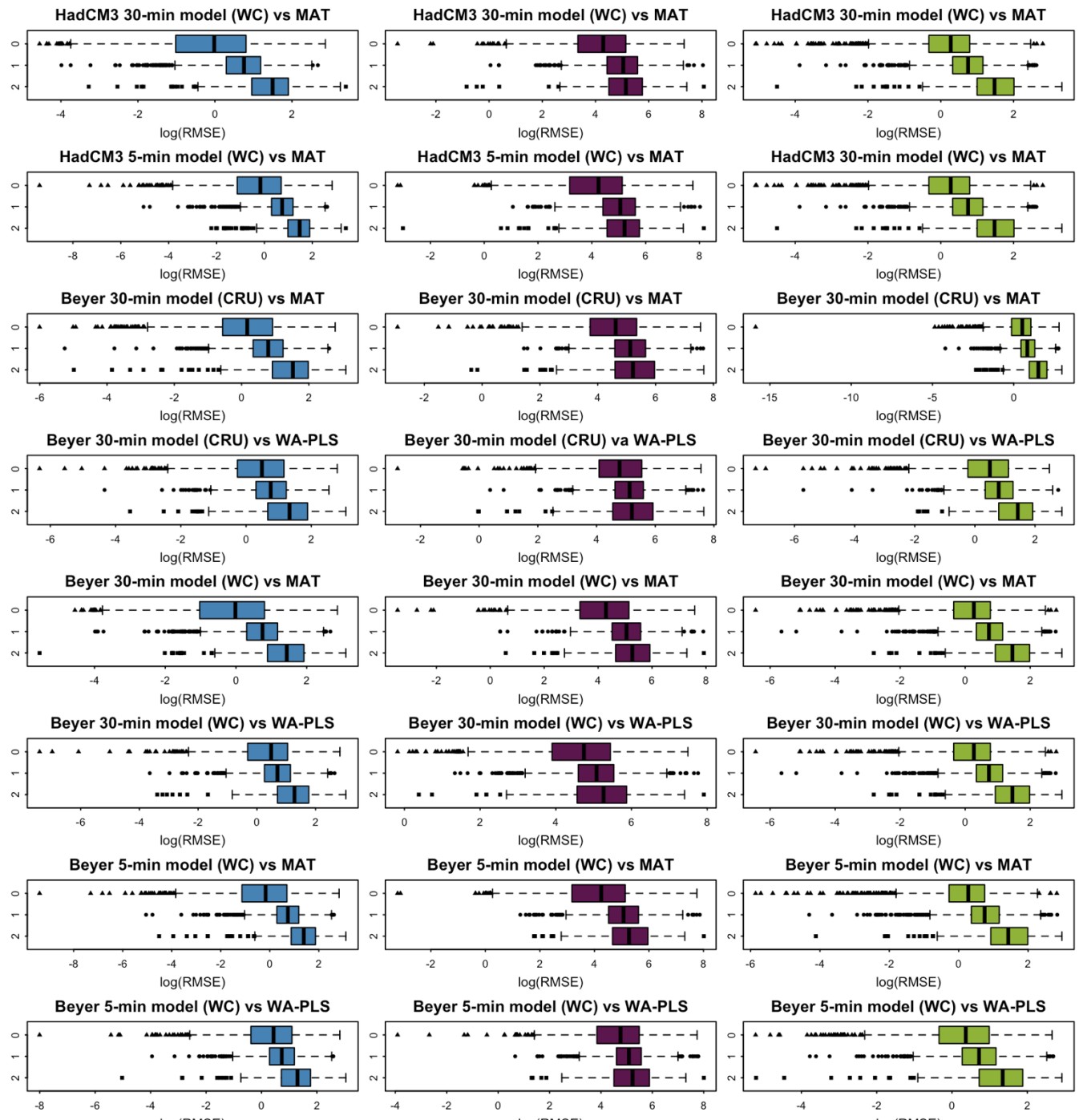

Figure 5. Boxplots of pair-wise log root mean square error (RMSE) results model-data comparisons of mean annual temperature (blue), mean annual precipitation (purple) and mean July temperature (green) from those representing the present (0), MIS 1 (1) and MIS 2 (2).

## 3.4 Exploring the most divergent time series

Observing the distribution of the data in Fig. 5 and Supplementary Figures S1 and S3, we decided to segment the highest 5% of RMSE values for each pair-wise model-data comparison for further investigation. We then amalgamated those that routinely fall into this category for each climatic variable, representing the most divergent time series of the overall dataset for the three parameters studied here (Appendix A Table A4). None of the individual records fall into the most divergent subset for all three variables studied, suggesting more extreme divergence is not related to any systematic issue in the model nor the proxy at

specific locations. We then produced 1000 bootstrapped samples (without replacement) of corresponding sample size, ascertaining whether the observed proportion of time series in this highly divergent subset is greater than expected by random chance (Appendix A Table A4).

To summarise, 44 records of mean annual temperature fall into the most divergent 5% of time series based on RMSE, of which

statistically significantly higher proportions of these than expected cover the present day and/or MIS 2, and/or are located in Asia, areas of high altitude and/or low roughness (Appendix A Table A4). For mean annual precipitation, only 21 records consistently fall in the top 5% based on RMSE, demonstrating higher inconsistency in pairwise model-data coherence between different model versions and methods of proxy reconstruction compared to the temperature variables (Appendix A Table A4). We found that, for this parameter, significantly higher proportions of these outliers are located in Asia and West North America

and/or in areas of high altitude and high roughness (Appendix A Table A4). Finally, for mean July temperature, 30 time series always fall into the most divergent 5%, significantly higher proportions of which date to the present and/or MIS 2, are located in Asia, areas of high altitude and/or areas of low topographic roughness than would be expected by chance (Appendix A Table A4).

Our results highlight that records spanning into MIS 2 consistently exhibit significantly higher proportions of divergent time series across all variables (Appendix A Table A4). This may specifically be a consequence of low $CO_2$ during MIS 2, which was not considered in LegacyClimate1.0, although this would mainly have an effect on moisture-related variables rather than temperature. Another potential source of divergence, leading to warmer reconstructions by proxies compared to the model outputs as well as significant deviations in precipitation, could derive from the geographic limits imposed on the

LegacyClimate1.0 proxies for the modern samples used to perform reconstructions. This is particularly problematic for the LGM as comparable signals should be present in the modern climate space within the limit defined (2000km around each site), which is likely unreasonable for some areas (e.g. northerly areas of Europe, see Fig. 1). Similarly, we find sites in Asia and higher altitude areas, where modern calibration data tend to be more limited, also have more divergent time series than expected given the sample size of this subset for all three variables (Appendix A Table A4). Sites in flatter areas exhibit significantly

higher proportions of divergent time series for annual and July temperatures than expected by random chance, whereas sites in higher roughness locations and West North America are more highly divergent than expected in precipitation (Appendix A

Table A4). Interestingly, we find that proxy records that capture the present day also occur in the most divergent subset more often than expected for annual temperature and precipitation, however this is because many of these records also span into MIS 2 (Appendix A Table A4).


## 4 Discussion

Increasing the spatial resolution of model time-series is often thought to be required to more accurately capture the climatic conditions of specific places at specific times. But what is the optimal spatial resolution for adequately detailing finer-scale signals? We tackle this question by testing the agreement between different model outputs and empirical reconstructions from

pollen proxies from the Late Quaternary for annual and July temperatures and annual precipitation. Ground-truthing modelled climate in this way is common, as proxies are considered to be the 'gold standard' for capturing more localised variations in climatic conditions in specific places. Our results highlight that further downscaling models via the delta method to much higher resolutions (5-minute) fails to *consistently* capture more of the climatic trend from pollen proxy records. Indeed, we were unable to demonstrate any statistically significant differences in model-data coherence between 30-min and 5-min model

resolutions in any subset of this large dataset. Overall, this implies that more downscaling may not always be the best solution, with relatively coarser simulations (i.e. 30-minute) providing a similarly adequate representation of past climatic trends in many scenarios, even in areas of topographic complexity. However, we stress that our take home message is not 'why bother', but that careful consideration should be required to determine when downscaling is important, given that coherence between proxy records and model outputs does not change significantly.


Regardless of resolution, we find that model-data coherence predictably decreases with age, with more divergent time series than expected by chance located in Asia, at higher altitudes and those capturing MIS 2. Annual precipitation and July temperature show consistently lower NRMSE than annual temperature, indicating good overall agreement between simulations and empirical reconstructions for these variables. Annual temperature shows low model-data convergence with greater

disparity between model outputs and methods of proxy reconstructions, as well as in certain contexts. Variability in coherence between regions likely relates to spatial variability in the performance of the simulations, the quality of modern reference datasets and proxy data employed, and the complexity in relationships between pollen and temperature tolerances in different geographic areas. Moreover, greater divergence at high altitudes and at older time scales may reflect limitation in the calibration with modern conditions, with reduced modern reference data at higher elevations and a lack of good analogues of

glacial/periglacial vegetations in the same areas as those in the past.

For this large-scale comparative analysis, we employed different de-biased and downscaled versions of the HadCM3 GCM output (Huntley et al. 2023) and Beyer *et al's.* (2020a) Late Pleistocene and Holocene climate simulations alongside harmonised pollen records from LegacyClimate1.0 (Herzschuh *et al.* 2023), providing corresponding estimates of three key

climatic parameters for comparison between time series. Whilst the LegacyClimate1.0 dataset provides an excellent

standardised and spatiotemporal expansive resource to address whether downscaling to higher resolutions is effective in capturing local climatic details, it is worth noting that, because the type of proxy records employed tend to capture pollen from a broad catchment, they may represent geographically wide averages of past climate. This could inherently make them more compatible with coarser-level model simulations, which also capture broader landscape rather than local-level trends. Future work should seek to expand systematic model-data comparisons on other types of harmonised proxies, as well as different climatic models and modern references, ensuring that the equivalent bioclimatic variables are being predicted by both sources.

Our results suggest that using statistical methods of downscaling simulated time series to much higher resolutions does not significantly improve the agreement between model output and pollen-proxy reconstructions, yet we note that there is a trade-off between enhancing spatial resolution and increasing potential error. Such error in a given location could either be caused by using too coarse a resolution on the one hand or by unreliable interpolation on the other. For this reason, there are likely to be many circumstances in which it is still better to use downscaled models (with caveats), particularly when variability within 30-min cells (~55km on each side) is important (e.g. Boisard et al. 2025). For example, the identification of conditions at specific locations within climatic extremes may be overlooked when using a model at a broader scale, such as at Late Pleistocene archaeological site Fincha Habera in the Bale Mountains of southern Ethiopia (Groos *et al.* 2021). Here, lower annual temperatures predicted by delta-downscaled models may better characterise the on-site environment than that also incorporating environmental trends in surrounding lower altitude landscape (Timbrell *et al.* 2022). Other methods of increasing model output, such as dynamical downscaling, may be better equipped for more localised applications, yet these are largely inaccessible for consumers of model output in fields like palaeoecology and archaeology where the computational costs are impractical. Overall, we present a streamlined pipeline for delta-downscaling climate model time series within the *pastclim* R package (Leonardi et al. 2023), and we have presented testing of downscaling using both HadCM3 model output (Huntley et al. 2023) and the product of Beyer et al (2020a) directly available within the package. We note that whilst the latter is not a direct output from a GCM, it is easily accessible for consumers (rather than producers) of model data, includes more sophisticated initial downscaling that takes advantage of a few runs of a high resolution GCM, and is likely to be used by others in the future as a starting point for further delta-downscaling.

## 5 Conclusion

Paleoclimatic proxies and climate models constitute two contrasting yet complementary sources of information on past climates. Demand for high-resolution climatic simulations that characterise landscape-scale heterogeneities come from the multitude of fields that employ ecological data, such as those that wish to map species distributions through time and space or quantitatively test hypotheses about the impact of climatic change and/or variability on various biological or behavioural phenomena. We show that downscaling via the delta-method fails to consistently capture more signal from temperature and precipitation proxy reconstructions, though model time series at both median (30-arc minutes) and fine-grained (5-arc minutes)

spatial resolutions characterise climatic variables in broadly similar ways to pollen proxies. Utilising model output for analyses of past climate therefore involves a careful balancing act between accentuating variations relevant to the study questions and
the potential introduction of error by unreliable interpolation.

**Code and data availability**

The workflow to downscale climate model outputs with the delta method has been made publicly available as functions in *pastclim*. Code and data relating to this analysis, as well as a vignette for downscaling in *pastclim*, was made available during the peer review of this article and can be found here: https://osf.io/duq3j/. The global downscaled models at 5-arc
minutes resolution are stored on Zenodo: https://doi.org/10.5281/zenodo.7828453.

**Author contributions**

Conceptualisation: LT, JB, MG, ES, AM; Data curation: LT, JB, MCh, AVP, AM; Formal analysis: LT, JB, AVP, MG, AM; Methodology: LT, JB, MG, MCh, AM; Software: LT, JB, MCo, ML, AVP, AM; Visualisation: LT, MCh; Writing – original draft preparation: LT, Writing – reviewing and editing: LT, JB, MCo, ML, MCh, AVP, MG, ES, AM.

**Acknowledgements**

LT, MCo and ES are supported by funding awarded by the Max Planck Society to the Human Palaeosystems Group. MCh is supported by the German Federal Ministry of Education and Research (BMBF) with the Research for Sustainability initiative (FONA) through the PalMod Phase III project (grant no. FKZ: 01LP2308B). ML and AM were funded by the Leverhulme Research Grant RPG-2020-317. AVP is supported by the Natural Environment Research Council grant number:
NE/S007164/1.

**Competing interests**

The authors declare that they have no conflict of interest.

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
