# Peer review of "More is not always better: delta-downscaling climate model outputs from 30 to 5-minute resolution has minimal impact on coherence with Late Quaternary proxies"

_Climate of the Past, 2024_

## Author Response (AR1)

**Response to reviewers for "**More is not always better: downscaling climate model outputs from 30 to 5-minute resolution has minimal impact on coherence with Late Quaternary proxies**"**

Reviewer 1:

**RC1:** This paper looks into the comparison between climate models and proxies and to what extent the differences between them could be reduced. The authors use statistical methods to increase the resolution of the model data to make it more comparable to proxy data, which represent local conditions. The conclusion is that even though the downscaled model data has more details the comparison with proxies is not really improved.

Considering the assumptions made and the methods used in the paper I wonder why anyone should expect an improvement of the model data. I suppose the paper can be a valuable contribution if these methods are commonly used in their part of the field. In any case, I think the authors should make it clear that their results apply to one particular type of statistical downscaling. It's not possible to draw any general conclusions about downscaling from these findings. Especially since the authors completely fails to mention dynamical downscaling.

Dynamical downscaling is known to improve the description of processes in the climate system and improve the description of local climate (e.g. Rummukainen, 2016). Dynamical downscaling is not very common within the field of palaeoclimate, but there are studies, e.g. Strandberg et al., 2011; Russo and Cubash, 2016; Velasquez et al., 2021; Strandberg et al., 2022; Strandberg et al., 2023.

Statistical downscaling is also known to improve local climate data and successfully minimize biases in climate models (e.g. Francois et al., 2020, Berg et al., 2022) Bias adjustment methods (also more advanced methods like quantile mapping) build on the assumption that the relationship between model and observations is constant. This works for the present and future (coming 100 years or so) climate because climate change is not that large. For palaeoclimates, however, you cannot expect this relationship to hold. You can't expect the model biases to be the same in the present climate as in the LGM or in the early Eemian. In a climate different from today, and with different topography the weather regimes are not the same as today – and therefore you can't expect the model biases to be the same as today and with different topography the weather regimes are not the same as today – and therefore you can't expect the model biases to be the same as today – and therefore you can't expect the model biases to be the same as today. If you in addition to these faulty assumptions use a very simplified method that only gives an offset of the model data, then I wonder why you at all expect your method to improve anything. Figure 2 clearly shows that your methods only slightly shifts model data. But you would like your method to also correct trends and variability.

**AC1:** We would like to thank the reviewer for drawing attention to our lack of discussion around dynamical downscaling, and for providing useful references. As the reviewer themselves suggests, dynamical downscaling is not very common within the field of palaeoclimate and associated fields (i.e. archaeology, palaeoecology etc.) who consume model outputs. This is because

this methodology is not accessible to researchers working with a large number of time steps due to the computational costs and time involved, particularly when exploring climatic variability over extended temporal or geographic spans. Yet tackling questions in archaeology, palaeoecology etc. often require finer levels of spatial resolution than typically provided by publicly available climatic model time series. We do not provide an overly extended discussion of these other methods of downscaling (i.e. dynamical downscaling), given they are not very relevant to our field, but add specific reference to them on Lines 71-75

"High resolution simulations of multiple time slices are often desired by consumers of model output yet difficult to obtain due to computational costs. For example, dynamical downscaling allows for the detailed description of processes in the climatic system and can improve the capturing of localised climatic conditions (Rummukainen, 2016; Strandberg et al., 2023), however this method is rarely applied in fields like palaeoecology and archaeology due to the computational costs, particularly when a large number of time steps are required."

And reiterate this point in the discussion on Lines 536-550:

"Our results suggest that using statistical methods of downscaling simulated time series to much higher resolutions does not significantly improve the agreement between model output and pollen-proxy reconstructions, yet we note that there is a trade-off between enhancing spatial resolution and increasing potential error. Such error in a given location could either be caused by using too coarse a resolution on the one hand or by unreliable interpolation on the other. For this reason, there are likely to be many circumstances in which it is still better to use downscaled models (with caveats), particularly when variability within 30-min cells (~55km on each side) is important (e.g. Boisard et al. 2025). For example, the identification of conditions at specific locations within climatic extremes may be overlooked when using a model at a broader scale, such as at Late Pleistocene archaeological site Fincha Habera in the Bale Mountains of southern Ethiopia (Groos et al. 2021). Here, lower annual temperatures predicted by delta-downscaled models may better characterise the on-site environment than that also incorporating environmental trends in surrounding lower altitude landscape (Timbrell et al. 2022). Other methods of increasing model output, such as dynamical downscaling, may be better equipped for more localised applications, yet these are largely inaccessible for consumers of model output in fields like palaeoecology and archaeology where the computational costs are impractical. Overall, we present a streamlined pipeline for deltadownscaling climate model time series within the pastclim R package (Leonardi et al. 2023), though we stress that careful consideration is required to select the optimal method and spatial resolution, based on the scope of the research question at hand."

We have also stressed the importance of testing the delta method as one of the most accessible methods of downscaling for consumers of palaeoclimatic model outputs on Lines 61-69:

"Models additionally offer much wider spatial coverage of the landscape that can be directly related to specific study sites and the palaeoclimatic differences between them. However, the integration of modelled climate with proxy data is not straightforward. For example, using simulations at a coarse resolution can produce biases when compared to on-site proxies due to the underlying complexity of the physical landscape, particularly in coastal and topographically diverse regions (Maraun and Widmann, 2018). Resultant differences can be in the order of several degrees for temperature and tens of percent for precipitation, which could lead to substantially different biome classifications and estimations of ecologies (Kottek et al., 2006). Such variations can have important implications for the diverse fields employing model output for the reconstruction of past and present species distributions, dispersal and extinction processes, and biogeographic patterns."

**RC2:** My point here is that the conclusions drawn in the paper are far too general. Statements like: "our results imply that downscaling to a very fine scale has minimal to no effect on the coherence of model data with pollen records." (l 28-29) are simply wrong. Your conclusions only apply to the methods used in this study, not all varieties of downscaling and bias adjustment.

**AC2:** We thank the reviewer for pointing out that some of our statements are too generalised. We have corrected this throughout the manuscript, for example on Lines 507-510:

"Our results highlight that further downscaling models via statistical methods to much higher resolutions (5-minute) fails to *consistently* capture more of the climatic trend from pollen proxy records. Indeed, we were unable to demonstrate any statistically significant differences in model-data coherence between 30min and 5-min model resolutions in any subset of this large dataset."

And added further specification that we are testing the delta method *specifically,* including in the title:

"More is not always better: delta-downscaling climate model outputs from 30 to 5-minute resolution has minimal impact on coherence with Late Quaternary proxies"

**RC3:** I think that the authors could be a bit more critical towards proxies. It's a bit much to call it "golden standard", and this comes from a modeller who is used to see all problems in models, and less so in proxies. Remember that proxies also have uncertainties. For example, Strandberg et al. (2011) come to the conclusion that the comparison between climate model and proxy data is mostly limited by the large errors bars in proxy data.

**AC3:** We have added further critique of proxies using the reference suggested by the reviewer, although we retain our stance that proxies are typically considered to be 'gold standard' by archaeologists, palaeontologists etc. when looking at climatic conditions at specific locations in the past:

Lines 99-102: "Proxies offer a more localised account of climate in certain places, yet they too can be associated with high degrees of uncertainty, arising from multiple sources. Nonetheless, determining model agreement with empirical reconstructions from proxies remains a widely applied method for ground-truthing downscaled climatic output."

Lines 107-113: "A recent meta-analysis by Laepple et al. (2023) found that studies in the Northern Hemisphere (where data are more abundant) have mixed results, suggesting potential areas of mismatch at local and regional scales. These authors suggest that shortcomings in both model simulations and proxy reconstructions may contribute to this divergence with models being less efficient at simulating local and regional temperature variability at relatively long timescales and methods of temperature reconstruction from proxies facing systematic deficiencies, though stronger emphasis is placed on the former. Strandberg et al. (2022) conversely suggest that comparisons between models and proxies are mostly limited by the large errors associated with proxy data."

**RC4:** I would also like you to think about the distribution between figures in the paper and the supplementary material. The paper doesn't include so many figures, and some of them are, to be honest, not that informative.

**AC4:** We have reworked all of the figures based on your specific suggestions (see below). Thank you.

**RC5:** At the same time the paper is quite heavy on reference to the supplementary. Perhaps you would like to lift something from the supplementary to the main text? And while you're at it rework some of the existing figures.

**AC5:** Thank you for this suggestion. We also apologise for the missing tables in the SOM. It was requested upon submission that that four tables from the manuscript be moved from the main text into the SOM due to CoP formatting issues. A new version of the SOM was submitted, including these 4 tables, but is unfortunate that this version was not shared with the reviewers nor uploaded online. We have however moved these large tables to an Appendix (Appendix A) so they are more easily accessible within the manuscript itself.

**RC6:** In conclusion, this paper has a very shallow description and discussion of downscaling and bias adjustment methods. This should be expanded. The conclusions should be reformulated to only apply to the methods used in the study, instead of all methods. If this is done, I think that the paper could be accepted (assuming that the methods are actually used in other projects). Otherwise I will recommend rejection.

**AC6:** Thank you for this summary. We believe we have sufficiently addressed all of your comments (see below) and would like to stress that accessible methods (i.e. that can be easily applied within a workflow, require manageable processing and accessible computational power) to downscale a large number of reconstructions are indeed very sought after in our field, who tend to be consumers of climatic model output as opposed to modellers.

**Comments**

**RC7:** L56-57 It could also be worth to mention that climate models also offer a picture that is also consistent across variables, thus giving a more complete picture of the climate.

AC7: We have amended Lines 59-61:

"Model output have the potential to overcome these shortfalls, providing tangible values for parameters such as temperature, precipitation, and a range of derived bioclimatic indices (e.g., Hijmans *et al.,* 2005), that are consistent across variables for a more complete account of climatic conditions."

**RC8:** L60 what do you mean by "observational data" here? Do you mean proxies? In that case, say so. Proxies and observations are different things. If you mean observations, explain why it is relevant to mention here. The rest of the paragraph is about proxies.

AC8: We have changed this to say 'proxy data' for clarity.

**RC9:** L63 "errors" Perhaps it's better to talk about "differences" since proxies also have errors.

AC9: Thank you for this suggestion; we have changed this to differences.

**RC10:** L71 "Different methods" -> "Different statistical methods". Otherwise you should also mention dynamical downscaling.

**AC10:** We have edited the manuscript accordingly and added more discussion about dynamical downscaling, as suggested on Lines 71-81:

"High resolution simulations of multiple time slices are often desired by consumers of model output yet difficult to obtain due to computational costs. For example, dynamical downscaling allows for the detailed description of processes in the climatic system and can improve the capturing of localised climatic conditions (Rummukainen, 2016; Strandberg et al., 2023), however this method is rarely applied in fields like palaeoecology and archaeology due to the computational costs, particularly when a large number of time steps are required. Most of the recently produced time series of palaeoclimate outputs have been downscaled from the native resolution of the models (usually in the order of 2 or 3 arc-degrees) to a higher resolution of 30 arc-minutes using

statistical methods (Fordham et al. 2017; Beyer *et al.* 2020a; Krapp *et al.* 2021; Zeller and Timmerman 2024; Mondanaro et al. 2025) as these approaches can be more easily applied to several time periods. Within statistical downscaling, different methods exist to increase the spatial resolution of model simulations; these include the delta method, generalised additive models (GAMs), and quantile mapping. These are all aimed at minimising biases in models, characterised as differences in statistical distributions between observed and simulated series."

**RC11:** Section 2.1 Here, I would like you to explain a bit more. It's difficult to follow what is done and in which order. Consider a more linear description, like GMC run, bias adjustment, downscaling etc. For example I don't understand what the Beyer et al simulation is. Is it a GCM run, a modification of the HadCM3 run or something else? Please also give some details about the HadCM3 run, for example regarding resolution and time span.

**AC11:** We now provide a detailed description of the output from Beyer et al. (2020), and the original HadCM3 model output (Huntley et al. 2022) we have subsequently added upon request from Reviewer 2 on Lines 129-170:

**"2.1 Climate models**

[revised manuscript text omitted]

RC12: L123-124 Is this the same simulation as in lines 112-113.

**AC12:** Yes, here we were referring to the Beyer et al. (2020a) output. We have adjusted the method sections to improve the clarity of our workflow (see above).

RC13: Eq. 1 Please explain what "DM", "sim", "raw" and "obs" denotes.

AC13: We have amended this section (Lines 172-190) as follows:

"For temperature variables, the bias in a geographical location x (a cell with a given latitude and longitude) is given by the difference between present-day observed  $T_{obs}(x, 0)$  and simulated  $T_{sum}^{\oplus}(x, 0)$  temperature, interpolated to the desired higher resolution grid via bilinear interpolation. Downscaled temperature  $(T_{sim}^{DD})$  in x at time t is thus estimated as

$$T_{sim}^{DD}(x,t) \coloneqq T_{sim}^{\oplus}(x,t) + \left(T_{obs}(x,0) - T_{sim}^{\oplus}(x,0)\right)$$

Precipitation is lower bounded by zero and covers different orders of magnitude across different regions compared to temperature. Multiplying rather than adding the bias correction is common when applying the delta method for precipitation, which corresponds to applying the simulated relative change to the observations (Maraun and Widmann, 2018). However, this method can therefore be hypersensitive in drylands, leading to overprediction of precipitation (and thus exacerbating the 'drizzling' bias of GCM). We have therefore adopted an additive approach for precipitation, analogous to the one used for temperature, with clamping within the range of observed maximum and minimum for current climate (see Beyer et al. 2020a). Like temperature, downscaled precipitation is estimated as

$$P_{sim}^{DD}(x,t) \coloneqq P_{sim}^{\oplus}(x,t) + \left(P_{obs}(x,0) - P_{sim}^{\oplus}(x,0)\right)^{*}$$

**RC14:** L161 Why do you use "bio01" here and "Tann" elsewere? Use a consistent terminology. I would prefer abbreviations like Tann instead of bio01, because they are easier to understand.

**AC14:** We use 'bio01' and 'Tann' etc. as this is how mean annual temperature are abbreviated in the climatic model and proxy dataset respectively. We retain bio01, bio12 and bio10 when describing the model output in the Methods and in Figures of the modelled climatic layers, however we use the full variable names (e.g. mean annual temperature) throughout the manuscript when discussing our results to ensure consistency.

We have added an additional sentence on Lines 216-221 explaining that these terms are equivalent variables:

"Our use of a single database reconstructing climate based on a single proxy reduces inter-site variability resulting from the type of data utilised and allows the generation of analogous climatic parameters with direct relevance to bioclimatic variables available in the Beyer et al. (2020a) model; Tann, Tjuly and Pann from LegacyClimate1.0 are the equivalent bioclimatic variables to bio01, bio10 and bio12 from HadCM3 GCM (Huntley et al. 2022) and Beyer et al. (2020a) model time series, which are standardly used in climatic modelling. "

Moreover, we have provided an account of the equivalent climatic variables extracted in Table 1, and have added an explanation of their abbreviations in Table 1.

"Table 1. Summary of the proxy records selected from the LegacyClimate 1.0 (Herzschuh *et al.*, 2023) and the model outputs (Beyer *et al.*, 2020a; Huntley et al. 2022) selected for analysis of mean annual temperature (bio01,  $T_{ann}$ ), mean July temperature (bio10,  $T_{july}$ ) and total annual precipitation (bio12,  $P_{ann}$ )."

**RC15:** L211-213 If this sentence is the only thing you write about Fig 2, why show it at all? I think it would be worth to describe also the differences between WAPLS and MAT.

**AC15:** We show Figure 2 as it visually captures the comparisons between time series that we are quantifying in this paper. We have added an additional sentence on Lines 259-261:

"Figure 2 highlights a sample of non-interpolated time series from proxy sites across the geographic span of the LegacyClim1.0 dataset, highlighting the coherence through time between different models and empirical reconstructions (WA-PLS and MAT) of the three climatic parameters (annual temperature, July temperature and annual precipitation)."

We do not think it is relevant to this paper to extensively describe the differences between the WA-PLS and MAT methods. These two state-of-the-art analytical methods have been commonly used in the field for over 3 decades, and there is ample documentation on how they work and how they perform in different situations. We feel that entering into technicalities would not add anything significant to the paper. However, and to guide interested readers, we have added three important references that correspond to extensive reviews of the field of pollen-based climate reconstructions that clearly highlight that the relative strengths and weaknesses of each of the methods (Sweeney et al., 2018; Birks et al. 2010; Chevalier et al., 2020). If the reviewer is referring here to the differences in *results* between WA-PLS and MAT, these are reported throughout Section 3, with limited variations between methods.

**RC16:** Fig 2 It's difficult to see the difference between the lines representing models. Consider using colours that are more different from each other, and to use dashes and dots to separate them even more.

**AC16:** We have made these suggested amendments by changing to a divergent colour scheme and using line representations to differentiate proxy from model time series in Figure 2.

---

## Author Response (AR2)

This version of the manuscript is much improved from the original. In particular, the figures have been greatly improved, and the addition of the Huntley et al. (2023) simulations brings the manuscript closer to its stated goals of investigating the impact of the downscaling of climate-model output on the comparison of that output with pollen-based reconstructions. I still think, however, that it is wrong to describe the Beyer et al. (2020a) data as "model simulations" (line 129) or "model output[s]" (line 149), because it was already downscaled/interpolated, and several steps removed from the actual Singarayer and Valdes (2010) etc. data. I fear that a naïve reader might think that there is therefore no benefit to the downscaling of actual model output when an easy-to-use data set can be found in an R package.

We thank to reviewer for acknowledging the improvements made to our paper.

Regarding our reference to 'model simulations' and 'model output', we standardised our terminology in response to this reviewer (Reviewer 2 RC5) who themselves suggested "model output" and "proxy reconstructions" were the correct terms to differentiate between the two sources of climatic data we are comparing. We have however added a comment on Line 578 to address the final point about the differences in the models utilised: "We note that whilst the latter [Beyer et al. 2020a] is not a direct output from a GCM, it is easily accessible for consumers (rather than producers) of model data, includes more sophisticated initial downscaling that takes advantage of a few runs of a high resolution GCM, and is likely to be used by others in the future as a starting point for further delta-downscaling." We have also edited Table 1 to further highlight the difference between the two model datasets.

Is there any benefit to using "real" (Huntley et al., 2023) model output as opposed to previously downscaled data? I agree that there may not be a statistically significant difference in performance, but Fig. 3 hints at a practical difference between the two (i.e the RMSEs for the Huntley et al.-based results appear lower than those for the Beyer et al.-based results).

We are unsure which figure the reviewer is referring to here, as Figure 3 demonstrates absolute bias values rather than RMSE (with no clear differences between Beyer et al. (2020a) and the HadCM3 GCM output in any of the figures). We note however that we included normalised RMSE (NRMSE) values for all of the comparisons, which is a standardised metric of coherence, and discuss the differences in NRMSE between the different models throughout the text (e.g. Lines 369-371: "In West North America, NRMSE is higher in the HadCM3 model outputs compared to that from Beyer et al. (2020a), with no differences between resolutions in the latter (when debiased using WC) and a slight improvement in coherence due to downscaling in the former"). However, there are no consistent 'practical' differences between the Beyer et al. (2020a) and the Huntley et al. (2023) models.

I think the authors did a good job of responding to reviewers' comments. However, I think three of their responses to my comments should also appear in the text. These are:

"AC1: We thank the reviewer for suggesting that we include a comparison with directly downscaled HadCM3 outputs. We have done so, using a model time series from Huntley et al (2022), which is an updated version of that used to generate Beyer et al. (2020a). The conclusions of our paper do not change... (p.14-15)"

We have added this sentence to Lines 119-122: "We applied a new suite of functions in the *pastclim* R package (Leonardi *et al.,* 2023) for delta-downscaling model output, including model-data comparisons with directly downscaled HadCM3 outputs from Huntley et al (2023), which is an updated version of that used to generate Beyer et al. (2020a), as well as the model time series from Beyer et al. (2020a)."

"AC3: Based on this, we do not believe that the take home message is 'why bother' but that careful consideration should be required to determine when downscaling is important, given that coherence between proxy records and model outputs does not change significantly. We understand that the reviewer is 'disappointed'... (p. 18)" (And I should point out that I was disappointed by the paper, not the results.)

We have added this sentence on Lines 536-539: "However, we stress that our take home message is not 'why bother', but that careful consideration should be required to determine when downscaling is important, given that coherence between proxy records and model outputs does not change significantly."

"AC20: We thank the reviewer for this suggestion. We have added the HadCM3 GCM from Huntley et al. (2022) to our analysis and find highly similar results with that of the Beyer et al. (2020a) output. Pertinently, we also find no statistically significant differences in coherence with proxy records between the HadCM3 GCM model output at 30-min and at 5-min resolution. We have retained Beyer et al (2020a) since it is an easily accessible product that includes more sophisticated initial downscaling that takes advantage of a few runs of a high resolution GCM, and it is likely to be used by others in the future (particularly consumers of climatic models) as a starting point for further delta-downscaling. (p. 22-23)"

This last comment exposes the authors' affinity for the data sets in the R package. They may be easy to use, but I don't think "easy" and "optimal" are the same thing.

We have added this sentence, and a caveat, to Lines 576-581: "Overall, we present a streamlined pipeline for delta-downscaling climate model time series within the *pastclim* R package (Leonardi et al. 2023), and we have presented testing of downscaling using both HadCM3 model output (Huntley et al. 2023) and the product of Beyer et al (2020a) directly available within the package. We note that whilst the latter is not a direct output from a GCM, it is easily accessible for consumers (rather than producers) of model data, includes more

sophisticated initial downscaling that takes advantage of a few runs of a high resolution GCM, and is likely to be used by others in the future as a starting point for further delta-downscaling."

Throughout the text, Huntley et al. (2022) should read Huntley et al. (2023).

We have made this correction throughout.